# ICF1-Syndrome-Associated *DNMT3B* Mutations Prevent De Novo Methylation at a Subset of Imprinted Loci during iPSC Reprogramming

**DOI:** 10.3390/biom13121717

**Published:** 2023-11-28

**Authors:** Ankit Verma, Varsha Poondi Krishnan, Francesco Cecere, Emilia D’Angelo, Vincenzo Lullo, Maria Strazzullo, Sara Selig, Claudia Angelini, Maria R. Matarazzo, Andrea Riccio

**Affiliations:** 1Department of Environmental Biological and Pharmaceutical Sciences and Technologies (DiSTABiF), Università degli Studi della Campania “Luigi Vanvitelli”, 81100 Caserta, Italy; ankitverma9079@gmail.com (A.V.); francesco.cecere@unicampania.it (F.C.); emilia.dangelo@unicampania.it (E.D.); 2Institute of Genetics and Biophysics (IGB) “Adriano Buzzati-Traverso”, Consiglio Nazionale delle Ricerche (CNR), 80131 Naples, Italy; varsh.pk@gmail.com (V.P.K.); vincenzo.lullo@igb.cnr.it (V.L.); maria.strazzullo@igb.cnr.it (M.S.); 3Department of Genetics and Developmental Biology, Rappaport Faculty of Medicine and Research Institute, Technion, Haifa 31096, Israel; seligs@technion.ac.il; 4Laboratory of Molecular Medicine, Rambam Health Care Campus, Haifa 31096, Israel; 5Istituto per le Applicazioni del Calcolo “Mauro Picone”, Consiglio Nazionale delle Ricerche (CNR), 80131 Naples, Italy; claudia.angelini@cnr.it

**Keywords:** ICF syndrome, DNMT3B, genomic imprinting, DNA methylation

## Abstract

Parent-of-origin-dependent gene expression of a few hundred human genes is achieved by differential DNA methylation of both parental alleles. This imprinting is required for normal development, and defects in this process lead to human disease. Induced pluripotent stem cells (iPSCs) serve as a valuable tool for in vitro disease modeling. However, a wave of de novo DNA methylation during reprogramming of iPSCs affects DNA methylation, thus limiting their use. The *DNA methyltransferase 3B* (*DNMT3B*) gene is highly expressed in human iPSCs; however, whether the hypermethylation of imprinted loci depends on DNMT3B activity has been poorly investigated. To explore the role of DNMT3B in mediating de novo DNA methylation at imprinted DMRs, we utilized iPSCs generated from patients with immunodeficiency, centromeric instability, facial anomalies type I (ICF1) syndrome that harbor biallelic hypomorphic *DNMT3B* mutations. Using a whole-genome array-based approach, we observed a gain of methylation at several imprinted loci in control iPSCs but not in ICF1 iPSCs compared to their parental fibroblasts. Moreover, in corrected ICF1 iPSCs, which restore DNMT3B enzymatic activity, imprinted DMRs did not acquire control DNA methylation levels, in contrast to the majority of the hypomethylated CpGs in the genome that were rescued in the corrected iPSC clones. Overall, our study indicates that DNMT3B is responsible for de novo methylation of a subset of imprinted DMRs during iPSC reprogramming and suggests that imprinting is unstable during a specific time window of this process, after which the epigenetic state at these regions becomes resistant to perturbation.

## 1. Introduction

Genomic imprinting is a process leading to mono-allelic and parent-of-origin-dependent expression of a subset of mammalian genes, generally based on differential DNA methylation of maternally and paternally derived alleles [1]. Differential DNA methylation of imprinted control regions (ICRs) is established in the female and male gametes, and 38 loci characterized by a germline-marked differentially DNA-methylated region (gDMR) have been identified in the human genome [2]. Each locus generally includes several imprinted genes regulated by the same gDMR through lncRNAs, histone modifications, insulators, and higher-order chromatin structure [1]. Secondary DMRs (sDMRs) are in near vicinity to gDMRs and acquire methylation during embryonic development in somatic cells [1]. Differential methylation of imprinted DMRs (iDMRs) is maintained in early embryonic cells despite the extensive genome-wide demethylation and remethylation waves that occur pre- and post-implantation [3]. Failure in establishing or maintaining imprinted methylation results in rare clinical disorders affecting growth, metabolism, endocrine, and neuro-behavioral functions [3].

Human induced pluripotent stem cells (iPSCs) mimic the implantation stage and thus can serve as a useful model to study the mechanisms underlying imprinting abnormalities at this developmental stage. However, frequent and recurrent variability of the DNA methylation patterns at certain imprinted loci may hamper the use of iPSCs as disease models [4]. A better understanding of the mechanisms controlling genomic imprinting in pluripotent stem cells is anticipated to improve the safety of iPSC-based applications. DNA methylation profiles of human iPSCs are globally similar to those of human embryonic stem cells (hESCs), and methylation levels of both these pluripotent cell types are consistently higher than those of somatic cells [4,5]. Indeed, cellular reprogramming involves a wave of de novo methylation necessary to suppress the genes driving differentiation, while promoters of pluripotency genes are fully demethylated [6,7]. The methylation patterns of several imprinted genes have been reported to be unstable in human iPSCs [4]. In particular, consistent hypermethylation of a subset of iDMRs is observed in iPSCs when compared to somatic cells.

DNA methyltransferase 3B (DNMT3B), one of the two de novo DNA methyltransferases, is highly expressed in human iPSCs. However, the role of DNMT3B in the methylation of iDMRs during the process of reprogramming human pluripotent stem cells is poorly investigated. During mouse spermatogenesis, Dnmt3b contributes with Dnmt3a and Dnmt3l to de novo methylation of the iDMRs [8]. In mouse embryonic stem cells (mESCs), Dnmt3b is recruited to gDMRs by interaction with the zinc-finger protein Zfp57 and its cofactor Kap1 to maintain imprinting methylation [9,10]. Since most gDMRs are located within gene bodies, these findings are consistent with the more general role of DNMT3B in ensuring intragenic methylation and fidelity of transcription initiation [11,12]. However, the role of Dnmt3b in imprinting maintenance during embryogenesis is debated. In mouse embryos, Dnmt3b has been shown to be dispensable for the maintenance of the gamete-derived methylation of gDMRs and rather appears to be involved in de novo methylation at many genomic regions during development, including imprinted sDMRs [13,14].

In humans, biallelic hypomorphic mutations in the *DNMT3B* gene cause immunodeficiency, centromeric instability, facial anomalies type 1 (ICF1) syndrome (OMIM #242860). ICF1 is an extremely rare immunological and neurological disorder characterized by hypogammaglobulinemia, T-cell dysfunctions, intellectual disability, and developmental delay [15,16]. ICF1 patients display DNA hypomethylation at subtelomeres and pericentromeric repeats, the latter being associated with decondensation of pericentromeric regions of chromosomes 1, 9, and 16 and related chromosomal rearrangements [17]. Additionally, non-repetitive genomic regions in ICF1 patient whole blood and derived lymphoblastoid cell lines display multiple changes in DNA methylation and transcription patterns [18,19]. Furthermore, pathogenic *DNMT3B* variants disrupt intragenic methylation and silencing of alternative and cryptic promoters in genes that regulate immune and neural cell function [20].

Although DNMT3B is highly expressed in human iPSCs, ICF1-patient-derived fibroblasts can be efficiently reprogrammed into pluripotent stem cells [21,22] similarly to *Dnmt3b*-deficient mouse somatic cells [23]. Notably, ICF1 iPSCs exhibit reduced levels of global methylation compared with normal iPSCs, indicating that DNMT3B contributes to de novo methylation during cellular reprogramming [21]. Consistent with these findings, we demonstrated that DNMT3B is required to methylate CGI-rich chromosomal domains in iPSCs derived from the fibroblasts of two different ICF1 patients [24]. We also showed that CRISPR/Cas9-mediated correction of *DNMT3B* mutations in isogenic iPSC lines enables the reacquisition of normal methylation levels at pericentromeric repeats [25] and the majority of hypomethylated regions. However, at the most severely hypomethylated regions in ICF1 iPSCs, which also display the highest increase in H3K4me3 levels and/or enrichment of CTCF-binding motifs, the epigenetic memory persisted, and hypomethylation was uncorrected [24]. The imprinted regions in the mammalian genome constitute a distinctive subgroup of loci that strictly maintain their DNA methylation throughout the de- and remethylation waves in early embryonic development [1]. However, during generation of iPSCs, the DNA methylation stability of these loci wavers. Here, using the unique platform of iPSCs carrying inactivating *DNMT3B* variants and their corrected counterparts, we investigated the role of DNMT3B in mediating the hypermethylation at imprinted loci during reprogramming of iPSCs.

## 2. Materials and Methods

### 2.1. Human iPSCs and Fibroblast Samples

ICF1 iPSC lines were derived from fibroblasts of two unrelated patients affected by ICF1 syndrome, subtype 1 (pG, male and pR, female) [22]. The isogenic clones, two from pG iPSCs (cG13, cG50) and two from pR iPSCs (cR7, cR35), in which *DNMT3B* mutations were corrected by CRISPR/Cas9-mediated editing, were described in previous studies [24,25]. As controls, we used 11 fibroblast-derived human iPSC lines, consisting of two internal iPSC lines derived from unaffected individuals (WT1, WT2) [24,25] and 9 samples whose DNA methylation profiles are publicly available (Appendix A). The iPSC lines were cultured on Geltrex LDEV-Free Reduced Growth Factor Basement Membrane Matrix (A1413302, Gibco, Thermo Fisher Scientific, Waltham, MA, USA) coated plates and maintained in StemMACS iPS-Brew XF, human medium (130-104-368, Miltenyi Biotec s.r.l. Bologna, Italy). Karyotype analysis validated the chromosomal integrity of WT and patient iPSC lines used in DNA and histone methylation studies. Fibroblast cells were grown in fibroblast media (DMEM media supplemented with 20% fetal bovine serum, 2 mM glutamine, 100 U/mL penicillin, and 100 mg/mL streptomycin) at standard conditions.

### 2.2. Methylation Array Analysis

Genomic DNA was extracted from iPSCs (passages 35 and 60 for pG and pR, 65, 40, 32, and 35 for cR7, cR35, cG13, and cG50, respectively) and parental fibroblasts using the Wizard^®^ Genomic DNA kit (A1120, Promega Corporation, Madison, WI, USA) following the manufacturer’s instructions. DNA was treated with sodium bisulfite, and bisulfite-converted DNA was subjected to genome-wide methylation profiling using Infinium Methylation EPIC 850K Bead Chip (Illumina Inc., San Diego, CA, USA). Fluorescence signal intensities were captured using Illumina HiScan SQ (Illumina Inc., San Diego, CA, USA). We analyzed the methylation array data in the R (v 3.6.1) environment using the ChAMP R package (v 2.20.1) [26]. First, we imported the raw IDAT files and the sample sheet using the champ.load function. Next, we set the quality control parameters as default and filtered out probes overlapping SNP positions, sex chromosomes, and detection *p*-value > 0.01. After that, we normalized the retained probes using the BMIQ (beta mixture quantile dilation) method. To increase the confidence of the detected differential methylation, we added to the internal controls human iPSC datasets from Gene Expression Omnibus (GEO) (Appendix A). Since public iPSC datasets consist of Infinium Methylation 450K arrays, we pre-processed them separately using the previously described procedure. Then, we considered only the shared CpGs with the EPIC array (850K) data. Next, we adjusted the combined matrix for batch effect using the Combat function. The normalized β-values were tagged with genomic features using the Illumina Manifest file after conversion to the hg38 genome using the liftOver R package (v 1.16.0). To identify differentially methylated CpGs, we converted the β-values to M-values using the beta2m function from the lumi R package (v 2.36.0). Then, to analyze individual CpGs, we computed the means and the standard deviations of control samples for each probe (using M-values). For the analysis at the genome level in iPSCs, we divided the genome into 500 bp bins and calculated the average methylation levels, β_ave_-value, in each 500 bp bin that contained at least 2 CpGs. Finally, we evaluated |ΔM| as the absolute difference between the control’s mean and individual samples’ M-values. Next, we considered differentially methylating those CpG probes or 500 bp regions that had |ΔM| larger than 3 times the standard deviation of the controls and with |Δβ| > 0.2 (where Δβ denotes the difference in the β-values or β_ave_-value of samples and median of the controls). Differentially methylated probes (dm-CpGs) or 500 bp regions (dm-500 bp bins) were further classified as hypermethylated if Δβ > 0.2 and hypomethylated if Δβ < −0.2. We removed the CpGs showing an absolute difference larger than 0.2 (in the β-value) between any control and the average of the controls. To detect wider hypomethylated regions, we combined contiguous dm-500 bp bins to form clusters containing at least 2 bins. The bins that did not belong to a cluster were called isolated bins. We used the approach described for dm-500 bp bins also to study differential methylation at specific genomic regions, such as iDMRs. The differentially methylated regions in pG or pR were defined as “recovered” in their respective corrected clones if |Δβ| < 0.2.

Finally, we annotated differentially methylated CpGs to various genomic features (i.e., exon, intron, intergenic, promoters, CDS, 5′UTR, 3′UTRs, CG intergenic, CpG islands, CpG shores, CpG shelves) using the annotatr R package (settings and hg38 reference database). Repetitive elements (namely LINES, SINES, LTRs, and Satellite) were annotated using “repeatmasker” file from UCSC (http://hgdownload.cse.ucsc.edu/goldenpath/hg38/database/rmsk.txt.gz) (accessed on 23 November 2021).

To evaluate the methylation profile of the pG and pR patients’ fibroblasts, we compared them with publicly available human dermal fibroblasts, as controls (Appendix A). Public methylation data of hESC samples were also used in the comparison between wild-type (WT) iPSCs and WT hESCs (Appendix A). We obtained methylation data of ICF1 patients’ and controls’ blood samples from [27]. As previously described for the iPSC dataset, we extracted the β-values from the raw IDAT files using the champ.load function and carried out quality control and filtering. After that, we performed BMIQ normalization employing the champ.norm function. Further, to take the common probes between the EPIC and 450K arrays, we used the combeArray function in minfi R package v1.44.0. For the analysis of the iDMRs, we downloaded the coordinates from http://www.humanimprints.net/ (accessed on 18 September 2021) and calculated the average methylation level for each region with at least 3 probes. Pyrosequencing analysis was conducted as previously described and using primers reported in our previous manuscripts [28,29].

### 2.3. ChIP-seq Datasets and Association with DNA Methylation Profiles

To investigate the differential enrichment of H3K4me3 and transcription factor CTCF at the hypomethylated imprinted loci, we used the ChIP-Seq datasets previously generated from WT iPSCs, ICF1 iPSCs, and corrected iPSCs [24], following the same processing steps and tools therein described. In addition, we evaluated the read count matrix at the iDMR intervals for the H3K4me3 immunoprecipitated replicates and input using the multiBamCov function from the BEDtools suite [30]. Then, the H3K4me3 enrichment at the iDMRs was calculated by normalizing the read count in IP and input by the sequencing depth and then computing the fold change (FC) = normalized read count of IP/normalized read count of input. ChIP assay to validate the ChIP-seq results was performed as previously described [24] and using the following primers for qPCR: ERLIN2:INTR6_F CAGCATCAATCACACGGGAC, ERLIN2:INTR6_R GAAGCCCCTCCATGTGTACT (113 bp, 60 °C); H19-IGF2_h3k4me4_F TTCACACTAGGGCCGAGATC, H19-IGF2_h3k4me4_R CAGGGGTCGTGAGGTATAGG (131 bp, 60 °C); H19-IGF2_CTCF_F GGTGACCCAGGACGTTTC; H19-IGF2_CTCF_R TGGCACAGAATCGGTTGTAAG (122 bp, 60 °C).

### 2.4. Data Visualization

Figures were generated with ggplot2 [31], ggpubR [32], pheatmap [33], and complexHeatmap packages in R [34]. To quantify the statistical significance of the methylation differences, we calculated the difference between ICF1(FC)–WT(FC) and corrected clone(FC)–WT(FC) for each CpG and applied the non-parametric paired Wilcoxon test with one-sided alternative and BH-FDR correction. Finally, we computed the effect size using the rank_biserial function from the effectsize R package between the mentioned differences [35]. We uploaded each sample’s methylation coverage tracks as β-values onto the UCSC genome browser session (https://genome.ucsc.edu/index.html) (accessed on 19 September 2021) [36]. For the sake of comparison, the tracks contain only methylation levels at common probes between EPIC and 450k arrays. Next, we also uploaded in the same session the bigWig coverage tracks of ChIP-seq (H3K4me3 and CTCF) replicates and input from [24]. We visually compared the hypomethylated iDMRs obtained in ICF1 iPSCs to hypomethylated regions (HMR) in WT human embryonic stem cells (hESCs) and hESCs in which *DNMT3B* was knocked out, as shown in the UCSC genome browser database. HMRs were defined as regions across the genome with low methylation levels identified using the MethPipe package [37]. We displayed the following HMRs available in MethBase track hub: WT hESCs HUES6 [38] and *DNMT3B* knockout hESCs (3BKO), late (17–22) passage [39].

### 2.5. Real-Time Quantitative PCR (RT-qPCR)

An amount of 1 µg of RNA derived from iPSCs was reverse transcribed using 100 ng of Random Primers (48190011, Invitrogen™, Carlsbad, CA, USA) and 100U of SuperScript™ II Reverse Transcriptase (18064022, Invitrogen™, Carlsbad, CA, USA) in a T100™ Thermal Cycler (Bio-Rad Laboratories, Hercules, CA, USA), according to manufacturer’s protocol (5′ at 65 °C, 2′ + 10′ at 25 °C, 40′ at 42 °C, and 15′ at 70 °C). RT-qPCR was performed using SsoAdvanced™ universal SYBR^®^ Green supermix (1725270, Bio-Rad Laboratories, Hercules, CA, USA) in a Bio-Rad iCycler, according to manufacturer’s protocols. Expression levels were normalized to the GAPDH gene by the ΔΔCt method. Primer sequences for gene expression are the following: H19F ATGACATGGTCCGGTGTGAC, H19R ATGTTGTGGGTTCTGGGAGC (162 bp, 62 °C); MEG3F GGGCATTAAGCCCTGACCTT; MEG3R CCTTGGGGAGGGAAACACTC (113 bp, 62 °C); MKRN3F GGAGGTGTTGGGAATATTGGG; MKRN3R AGAAAGGGAAGGGGAGTAGG (181 bp, 62 °C); GAPDHF ACATGTTCCAATATGATTCCA, GAPDHR TGGACTCCACGACGTACTCAG (162 bp, 62 °C).

### 2.6. Statistical Analysis

Specific statistical tests are described in figure legends and the Materials and Methods section. We adjusted the *p*-values with the Benjamini–Hochberg method. The data analyzed using RT-qPCR are presented as mean ± SD from independent triplicates, each amplified twice. Statistical analyses were performed using the one-sided two-sample Student’s *t*-test; (*) *p*-value < 0.05, (**) *p*-value < 0.01, (***) *p*-value < 0.001, (****) *p*-value < 0.0001. Statistical significance was calculated by comparing each ICF1 or corrected iPSCs to each WT (WT1 and WT2) iPSCs separately, and the least significant values were reported in the corresponding figures. The statistical significance of the methylation differences between ICF1 iPSCs and corrected clones from the median of controls was calculated using a non-parametric two-sample two-sided Wilcoxon test with BH-FDR correction.

### 2.7. Data Access

Our Illumina EPIC array raw and processed data, including iPSCs and fibroblast samples, are deposited in the GEO under accession number GSE198705. ChIP-seq raw and processed data are publicly available under accession number GSE197925 [24].

## 3. Results

### 3.1. De Novo Gain of Methylation at Imprinted Loci during iPSC Reprogramming Is Affected by DNMT3B Loss of Function and Is Not Restored following Gene Correction

Our previous study of CpG methylation in iPSCs derived from two unrelated ICF1 patients (pG and pR) and their corresponding DNMT3B-corrected isogenic clones (cG13, cG50, and cR7, cR35) demonstrated that the majority of hypomethylated genome-wide CpGs in patient iPSCs were restored to normal levels following DNMT3B correction [24]. However, the analysis of specific genomic regions characterized by intermediate DNA methylation levels and a certain degree of epigenetic instability in iPSCs, such as imprinted loci [4], was limited by the relatively low coverage and low number of available control samples subjected to whole-genome BS-seq. To overcome this limitation, here we used genome-wide methylation arrays and compared the methylation profiles of ICF1 iPSCs and corrected clones with those of 11 control human iPSC lines, two of which were generated in-house and nine were elsewhere published (Appendix A) [40,41,42].

We identified differentially methylated CpGs consisting of 20,234 hypomethylated and 402 hypermethylated CpGs in pG and 24,838 hypomethylated and 998 hypermethylated CpGs in pR. Among them, 17,004 hypomethylated and 224 hypermethylated CpGs were common to both pG and pR iPSCs. Further, out of 34,048 500 bp bins, we identified 940 and 1285 differentially methylated (dm) bins in pG and pR, respectively (Appendix A). As for individual CpGs, most of the dm-500 bp bins were hypomethylated in pG and pR (Appendix A). The majority of the dm-500 bp bins in pG and pR recovered normal methylation levels in the corrected clones cG13 (84.2%) and cG50 (70.8%) and cR7 (91.6%) and cR35 (75%), respectively (Appendix A). Among the shared dm-500 bp bins that were affected in both pG and pR (*n* = 767), 57.3% (*n* = 440) recovered the proper methylation status (|Δβ| < 0.2) in all four corrected iPSCs (Appendix A).

As reported in our previous study [24], we observed that some hypomethylated bins clustered into the same genomic regions to form larger hypomethylated domains. Specifically, here we identified 48 clusters containing 109 dm-500 bp bins in pG and 78 clusters containing 169 dm-500 bp bins in pR (Appendix A). For example, the regions spanning the *PCDH* gene cluster were hypomethylated in ICF1 iPSCs and fully recovered their DNA methylation levels in corrected iPSCs (Appendix A). In contrast, the major hypomethylated cluster of the *TNXB* gene in ICF1 iPSCs overlapped with a CpG island that failed to recover normal DNA methylation in the corrected clones (Appendix A).

Next, we annotated the dm-CpGs to various classes of genomic elements and found that the majority of them recovered normal DNA methylation levels upon DNMT3B correction with similar efficiency (Appendix A, Appendix A). However, the methylation level at CpG islands was restored less efficiently, while satellite repeats showed the highest percentage of DNA methylation recovery (Appendix A). Compared to the hypomethylated probes, the distribution of hypermethylated regions across genomic elements, and their rescue in the corrected clones, was more heterogeneous between the two ICF1 patients, although they overall significantly overlapped (shuffle test; *p* < 0.000001). Overall, the genome-wide methylation results obtained using arrays completely recapitulated the findings previously achieved by WGBS sequencing, thus providing a reliable tool to study methylation profiles at specific regions in the genome. To gain insights into the role of DNMT3B in methylating imprinted regions during iPSC reprogramming, we focused on a previously published list of 50 germline and somatic human iDMRs (Appendix A) [2]. We first filtered out those containing < 3 CpGs, retaining 42 out of the 50 iDMRs. After further removing VTRNA2 from this list due to its polymorphic nature [43], we were left with 41 iDMRs to analyze. Next, we calculated the average methylation for each iDMR and identified the differentially methylated iDMRs in ICF1 iPSCs compared to WT iPSCs by applying the strategy already used for the 500 bp bins. Our analysis showed three groups of iDMRs based on their DNA methylation levels in control iPSCs (Figure 1A).

A subset of iDMRs was fully or partially hypermethylated, and a second group displayed the intermediate methylation level (approximately 50%) expected for imprinted control regions, while a smaller subset showed low methylation levels. When compared to control iPSCs, a subset of hypermethylated imprinted loci showed lower levels of methylation in both ICF1 pG and pR iPSCs, indicating that DNMT3B contributes to their gain of methylation during reprogramming of WT iPSCs (Figure 1A, Appendix A and Appendix A). In particular, the gDMRs *H19/IGF2*, *DIRAS3_ex2*, and *HTR5A* and the sDMRs *IGF2_ex9* were severely hypomethylated, and *PEG13* was moderately hypomethylated in both pG and pR iPSCs (Figure 1A,B). Among the hypermethylated iDMRs, we found that the *GPR1-AS*, *RB1*, *PEG3*, and *WRB* gDMR and the *SNRPN:int1* and *SNRPN:int2* sDMRs were not perturbed by DNMT3B loss of function.

A few imprinted regions showed reduced DNA methylation levels in both pG and pR but were significantly hypomethylated only in one of the two ICF1 iPSC lines (*ERLIN2* gDMR, *MKRN3* sDMR, and *ZNF331_2* gDMR). Some other imprinted DMRs were hypomethylated in pG (*MEG3* sDMR, *L3MBTL1* gDMR, and *NNAT* gDMR), while they were not affected at all in pR iPSCs (Figure 1A,B and Appendix A). The *IGF2R*, *IGF2:alt-TSS*, and *ZNF597_1* gDMRs were also hypomethylated, but less than three CpGs were included in the methylation array. In addition, variable levels of hypomethylation of some DMRs (e.g., *L3MBTL1*, *NNAT*, *MCTS2P*, and *PPIEL*) were observed in several of the control iPSCs, consistent with the reported instability of imprinting during iPSC reprogramming [4] (Appendix A).

We further assessed the methylation level of CpGs at hypomethylated iDMRs following CRISPR/Cas9 correction of *DNMT3B* mutations in the patient iPSCs. Compared to the bulk of hypomethylated CpGs throughout the genome that regained normal DNA methylation levels, as exemplified by the *PCDH* gene cluster (Figure 1C; Appendix A), we observed that most hypomethylated iDMRs were resistant to remethylation in all four corrected clones (Figure 1C; Appendix A). CpG methylation was partially restored at some imprinted loci, as at *IGF2:ex9* sDMR, or recovered in only one of the two corrected clones of each ICF1 iPSCs, as at *DIRAS3:ex2*, *MEG3* gDMR, and *MKRN3* sDMR (Figure 1A; Appendix A).

To evaluate the DNA methylation changes in the imprinted DMRs during cell reprogramming, we determined their DNA methylation level in the original parental fibroblasts of the two ICF1 patients compared to WT fibroblasts. We observed that the subset of imprinted loci that was hypermethylated in WT iPSC showed approximately 50% methylation in WT fibroblasts, with a few exceptions (e.g., *GPR1-AS, NNAT:TSS*, *and DIRAS3:EX2*), indicating that these DMRs undergo partial or full de novo methylation during the process of cellular reprogramming to a pluripotent state (Figure 2A, left panel). The hypermethylation profile of iDMRs is consistent with that observed in hESCs and is clearly distinguished from that of fibroblasts, confirming that it is characteristic of the pluripotent state (Figure 2B). Overall, DNA methylation levels appeared closer to intermediate levels (50%) in fibroblast cells, suggesting that the imprints are more stable in somatic cells than in pluripotent stem cells. Consistently, the methylation levels of iDMRs were approximately 50% in blood samples from control individuals (Figure 2A, right panel).

In both ICF1 patient fibroblast and blood samples, some iDMRs showed DNA methylation defects that were less pronounced when compared to iPSCs (Figure 2A). Other iDMRs (e.g., *DIRAS3*, *H19/IGF2*, *PEG13*, *PPIEL*, and *MCTS2P*) were hypomethylated exclusively in the patient iPSCs. Furthermore, methylation levels were generally lower at several secondary iDMRs (*ZNF597_TSS*, *MAGEL2*, *NDN-TSS*, *SNRPN:alt-TSS*, *SNRPN:int1*, and *SNRPN:int2*) in patient fibroblast and blood samples but moderately affected in patient iPSCs compared to their controls (Figure 2A). Pyrosequencing analysis of DNA methylation level at a representative subset of imprinted DMRs in iPSC and fibroblast samples further validated the results of the methylation array (Appendix A).

Taken together, our results suggest that the mutated DNMT3B in ICF1 cells interferes with de novo methylation of a subset of imprinted DMRs during cellular reprogramming into iPSCs, and gene correction in ICF1 iPSCs cannot restore the methylation levels of iDMRs to those of control iPSCs.

### 3.2. The H3K4me3 Mark Is Abnormally Enriched at Hypomethylated Imprinted DMRs

Loss of DNMT3B function can lead to increased deposition of histone H3K4 trimethylation at hypomethylated loci, which in turn can alter gene expression patterns [20,24,25]. To evaluate whether the H3K4me3 mark levels influence the ability of the corrected DNMT3B protein to remethylate the iDMRs, we exploited the ChIP-seq data previously obtained [24]. We observed a concomitant increase in H3K4me3 levels at the hypomethylated imprinted loci compared to those unaffected in ICF1 iPSCs (Figure 3A). The abnormal H3K4me3 levels persisted at the imprinted DMRs resistant to remethylation in the corrected clones (*HTR5A:TSS DMR* and *ERLIN2:Int6 DMR*) (Figure 3A, Appendix A), while were reverted to the normal state at those loci which regained normal DNA methylation levels in the corrected clones (*IGF2:ex9* and *DIRAS3:ex2*; Appendix A). This observation further confirms the role of abnormally high H3K4me3 levels in inhibiting the recruitment of the corrected DNMT3B protein to the affected DMRs and their remethylation.

At the hypomethylated *H19/IGF2* gDMR, the H3K4me3 enrichment was comparable between WT, patient, and corrected iPSCs (Figure 3B and Appendix A), and we did not observe a further significant increase in this histone mark as a consequence of DNMT3B loss of function. Because the transcription factor CTCF plays a key role in regulating the DNA methylation pattern at the *H19/IGF2* imprinted locus [44], we took advantage of the ChIP-seq data generated in patient and corrected iPSCs [24] to explore the CTCF binding profile at this locus. Indeed, we found an aberrant increase in CTCF binding at the *H19/IGF2* locus in patient iPSCs compared to WT iPSCs that also persisted in the corrected clones (Figure 3B and Appendix A). This finding suggests that bound CTCF may represent an additional factor that negatively affects the recruitment of the corrected DNMT3B protein to this DMR. Notably, the *H19/IGF2* gDMR is similarly hypomethylated in *DNMT3B* knockout (KO) hESCs relative to WT hESCs, supporting the notion that this genomic region is a direct target of DNMT3B protein during an early developmental time window (Figure 3B).

The increased H3K4me3 levels prompted us to investigate the expression of the imprinted genes associated with abnormally hypomethylated DMRs in ICF1 iPSCs. Because in *Dnmt3b* knockout mouse embryos the hypomethylated imprinted genes *H19*, *Meg3*, and *Mkrn3* displayed transcriptional upregulation [13], we assayed the expression of these hypomethylated imprinted genes in ICF1 iPSCs and corrected clones by RT-qPCR. We observed that the transcription of the *MEG3* gene, which was hypomethylated exclusively in pG iPSCs, was activated in pG iPSCs compared to control iPSCs, and its deregulated expression was restored to normal in cG13 and cG50 clones (Figure 3C). Conversely, the *MKRN3* gene was hypomethylated and was upregulated only in pR iPSCs. This aberrant transcription was partially or entirely reverted to normal levels in cR7 and cR35 clones, respectively. The *H19* gene was transcriptionally derepressed in both pG and pR iPSCs and their corrected clones.

Altogether, our data support that DNMT3B has a role in the epigenetic regulation of a subset of iDMRs. This regulation is associated with H3K4 trimethylation levels and occasionally with transcription levels, as well as CTCF binding as in the case of the *H19/IGF2* locus. Following correction of *DNMT3B* mutations, the methylation level is restored to WT levels at the iDMRs where the abnormally high H3K4me3 levels were reverted back to normal levels.

## 4. Discussion

Human imprinting defects mostly arise due to a failure in maintaining differential methylation of the allelic gDMRs during early embryogenesis [3]. So far, familial cases of imprinting disorders have been useful to identify several trans-acting factors involved in methylation maintenance. However, the majority of imprinted methylation defects arise sporadically, likely due to stochastic errors, environmental influence, or multigenic variants, making the clinical cases less informative. Also, species-specific differences in imprinting control render the study of human genetic variants in animal models very complex [45]. For these reasons, cellular models represent a valuable system for investigating the mechanisms underlying imprinting maintenance in the human species.

Human iPSCs are considered as a powerful tool for disease modeling. However, several studies have reported epigenetic instability arising during reprogramming of somatic cells into iPSCs [4,5]. Comparative analyses indicated that the DNA methylation patterns of several iPSCs, regardless of their source tissue, were globally similar to each other, as well as to those of human embryonic stem cells (hESCs) [4]. However, iPSC and hESC methylation profiles exhibited certain epigenetic differences, which were caused by random aberrant hypermethylation at early passages. Prolonged culture reduced the heterogeneity of the DNA methylation profiles among the iPSC population and the differences compared with hESCs [6,7]. While naive PSCs are genome-wide hypomethylated and resemble pluripotent cells in vivo [46], the global persistence of DNA methylation in in vitro cultured human pluripotent stem cells (hESCs and iPSCs) has been proven essential for their survival, as 5AzaC treatment or *DNMT1* knockout leads to rapid cell death [37,47,48].

Genomic imprinting is noticeably affected by DNA methylation heterogeneity in human ESCs and iPSCs [4,49,50]. In particular, *H19*, *IGF2*, and *MEG3* have often been found to be unstable in human ESCs [49]. *MEG3*, *H19*, *DIRAS2*, and *ZIM2/PEG3* were also found to be aberrantly hypermethylated in several iPSC lines [4,51]. In the present study, we took advantage of our cellular model based on iPSCs derived from ICF1 patients carrying pathogenic *DNMT3B* variants to address the role of DNMT3B in regulating imprinting during reprogramming. In order to quantify DNA methylation levels of imprinted domains at higher precision than those obtained by our previous BS-seq data, we chose a DNA-methylation-array-based platform. Despite covering a lower number of CpGs, this platform provides more accurate information regarding small DNA methylation differences at each covered CpG within the imprinted loci. The whole-genome methylation analysis of mutant and isogenic iPSC lines, in which *DNMT3B* mutations were corrected, provided a comprehensive picture of the specific methylation activity of DNMT3B at 41 imprinted regions throughout the human genome.

Our findings consolidated the notion that reprogramming of fibroblasts into pluripotent stem cells involves an abnormal gain of methylation at several imprinted loci. In addition *to H19/IGF2*, *MEG3*, *DIRAS3*, and *PEG3* DMRs, we observed consistent hypermethylation of *SNRPN:Int1 DMR2*, *WRB*, *RB1*, *ERLIN*, and *MKRN3* in WT iPSCs when compared to WT fibroblasts. A lower gain of methylation was found at *PPIEL*, *HTR5A*, *IGF2:Ex9*, *MCTS2P*, *ZNF331*, and *SNRPN:Int1* (Appendix A). Overall, we observed a 50% methylation level more consistently in WT fibroblasts and blood samples, compared to iPSCs, confirming that—differently from pluripotent stem cells—genomic imprinting is rather stable in somatic cells.

Notably, within all iDMRs, the subset of hypermethylated iDMRs in WT iPSCs is the group most highly affected by DNMT3B deficiency in ICF1 iPSCs (Figure 2A and Appendix A). Several imprinted DMRs, such as, for example, *ERLIN2:Int6*, *HTR5A:TSS*, and *IGF1R:Int2*, were hypomethylated in both patient iPSCs and the parental fibroblasts, when compared to control cells, while a few other iDMRs were affected only in the patient iPSCs (*DIRAS3*, *H19/IGF2*, and *MCTS2P*), indicating that DNMT3B is responsible for the gain of methylation during cell reprogramming. The profile of the methylation defects at iDMRs was more similar between ICF1 patient fibroblasts and blood samples than to that of patient iPSCs, and included several secondary iDMRs (*SNRPN:Int1*, *SNRPN:Int2*, and *NDN*).

Despite deficient DNMT3B activity, we observed that several imprinted DMRs are efficiently methylated during reprogramming of ICF1 iPSCs from fibroblasts (*RB1, SNRPN:Int2, PEG3, WRB, SNRPN:Int1*) and are thereby protected from DNMT3B loss of function. This may be explained by specific sequence features and/or by molecular factors promoting the targeting of the residual DNMT3B activity and/or of DNMT3A protein to these iDMRs.

The DNA methylation profiles at iDMRs are consistent with our previous findings at subtelomeric regions, where the loss of DNA hypomethylation persisted at certain subtelomeres after reprogramming into ICF1 iPSCs, whereas others were partially or fully remethylated [25]. However, while at subtelomeres the severity of hypomethylation in ICF1 fibroblasts influenced the ability to regain methylation during reprogramming, this was not observed at the imprinted loci. Indeed, *SNRPN:Int1* and *SNRPN:Int2*, which show the lowest methylation levels in patient fibroblasts compared to controls, are readily methylated in derived iPSCs.

The genomic position of individual iDMRs might also influence their sensitivity to DNMT3B loss of function, because nearby DMRs were differently perturbed in ICF1 iPSCs, as in the case of *DIRAS3:Ex2* and *DIRAS3:TSS* gDMRs. This supports the notion that DNMT3B may have a stronger affinity for certain imprinted control regions compared to others. The DNA methylation level of the two imprinted DMRs DIRAS3:Ex2 and *DIRAS3:TSS* was also differently affected in *DNMT3B*-knockout human ESCs, suggesting that only *DIRAS3:Ex2* DMR is presumably a direct target of the DNMT3B protein in pluripotent stem cells [39]. It is interesting to note that in this case the hypomethylated iDMR is localized in the gene body, whereas the unaffected iDMR is at the TSS, in line with the evidence that DNMT3B preferentially methylates intragenic regions rather than promoters. However, we also observed hypomethylation at the TSS in a few cases, such as, for instance, *HTR5A_TSS*, suggesting that multiple factors act to modulate the affinity of DNMT3B for different iDMRs.

Some iDMRs hypomethylated in ICF1 iPSCs are already known to be specifically regulated by Dnmt3b activity in mice. In mouse embryos, Dnmt3b is necessary for de novo methylation of sDMRs during development, while it is dispensable for maintaining methylation at gDMRs [14]. We did not observe preferential hypomethylation of sDMRs compared to gDMRs nor of paternally compared to maternally methylated DMRs in the ICF1 iPSCs. However, the *H19, Meg3*, and *Mkrn3* iDMRs, which were hypomethylated in *Dnmt3b* −/− mouse embryos [13], were also affected in ICF1 iPSCs. Consistent with the findings in *Dnmt3b* knockout mice, where the transcription of hypomethylated imprinted genes *H19*, *Meg3*, and *Mkrn3* was upregulated [13], the methylation defect in ICF1 iPSCs of *H19, MEG3*, and *MKRN3* iDMRs was specifically associated with a slight increase in their associated transcripts. The RNA level of the other hypomethylated imprinted genes was unchanged in the ICF1 iPSCs, possibly because of their tissue specificity or post-transcriptional compensatory mechanisms [52].

A major finding of our study is that the hypomethylated imprinted DMRs demonstrate a general resistance to remethylation upon restoration of DNMT3B activity. In contrast, the majority of the hypomethylated CpGs of the genome regain normal methylation in the corrected clones [24]. Among the various genomic elements, hypomethylated CpG islands were more resistant to reversal of the defective epigenetic state acquired in the ICF1 iPSCs. In addition, the hypomethylated imprinted DMRs were found to be abnormally enriched with the H3K4me3 mark [24]. It was shown that high CpG density and H3K4me3 enrichment of the non-imprinted allele are characteristic features of imprinted DMRs [3]. Thus, it is likely that the resistance of the hypomethylated iDMRs to restore DNA methylation in rescued ICF1 iPSCs is due to the acquisition of the characteristics of the non-imprinted allele at both parental chromosomes in the absence of DNMT3B activity. Consistent with this suggestion, reintroduction of ZFP57 into mouse *Zfp57*-null ES cells did not result in reacquisition of a DNA methylation imprint, indicating that genomic imprinting cannot be recovered in somatic cells [53].

From a mechanistic point of view, the abnormally high H3K4me3 levels represent a molecular barrier that inhibits the recruitment of the corrected DNMT3B protein. Indeed, we previously demonstrated that pharmacological reduction of this barrier partially improved methylation recovery at hypomethylated subtelomeres resistant to DNA methylation correction [25]. Interestingly, the molecular barrier at *H19* gDMR is presumably achieved by aberrant binding of CTCF in ICF1 iPSCs, and it persists in corrected clones. CTCF does not abnormally bind at the other hypomethylated imprinted DMRs. Thus, additional factors are predicted to interfere with the restoration of DNA methylation levels to the normal state, such as, for example, TET enzymes and/or methylation-sensitive transcription factors. Further studies will be required to characterize the contribution of these additional factors to persistent hypomethylation at iDMRs in ICF1 iPSCs.

Overall, this study confirms that maintenance of genomic imprinting is unstable during reprogramming of pluripotent stem cells and shows that more stability and resistance to perturbation are present after the pluripotency state has been achieved. Moreover, our findings demonstrate the role of DNMT3B in the de novo methylation of a subset of imprinted DMRs in human iPSCs during cellular reprogramming. Awareness of this caveat in iPSCs and understanding the molecular mechanisms responsible for this phenomenon are steps towards generation of iPSCs that will serve clinical needs in a safe and secure manner.

## Figures and Tables

**Figure 1 biomolecules-13-01717-f001:**
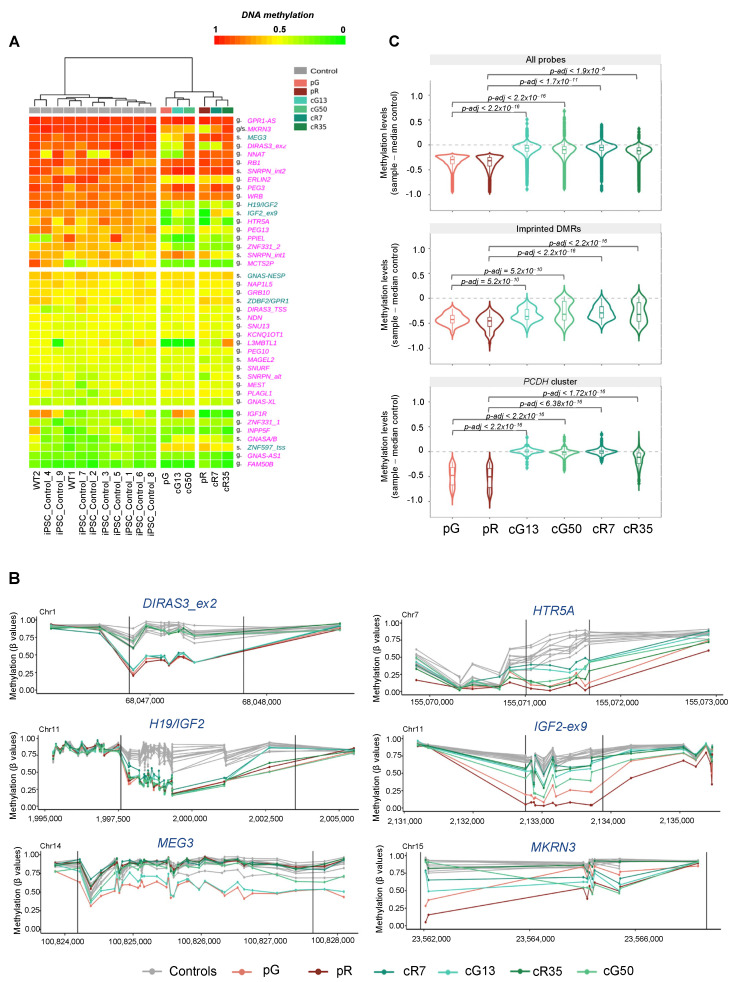
(**A**) Heatmap depicts average methylation at imprinted loci (*n* = 41) in ICF1 patient iPSCs, their corrected counterparts, and control iPSCs (WT1, WT2, internal controls, and nine wild-type (WT) iPSC samples taken from external databases). The iDMRs were divided into three groups based on the mean methylation levels in control iPSCs (>60%, 40–50%, and <40%). A color scale of DNA methylation level is shown. Maternally methylated germline (g) and secondary (s) DMRs are in pink while paternally methylated germline (g) and secondary (s) DMRs are in blue. (**B**) Line charts depict the methylation profiles at differentially methylated loci in both pG and pR (≥3 CpGs inside DMRs), corrected iPSC lines, and controls (*n* = 11). Dots on the line show the position of CpGs. Two black lines within the plot represent the span of the imprinted DMR. The x-axis represents genomic coordinates in and around the iDMRs, while the y-axis shows methylation levels (β-values). (**C**) Violin plots of the methylation levels in ICF1 and corrected iPSC samples, represented as a difference between pG, cG13, and cG50 or pR, cR7, and cR35 and the median of controls (*n* = 11), at all hypomethylated CpGs (pG *n* = 20,234, pR *n* = 24,838; upper panel), the subset of hypomethylated CpGs of iDMRs (pG *n* = 87, pR *n* = 69; middle panel), and the hypomethylated CpGs included in the *PCDH* gene cluster (pG *n* = 208, pR *n* = 306; bottom panel). Boxes inside the violin are interquartile ranges with the median line. *p*-adjusted values represent the BH-corrected *p*-values obtained from a two-sample Wilcoxon test with two-sided alternatives.

**Figure 2 biomolecules-13-01717-f002:**
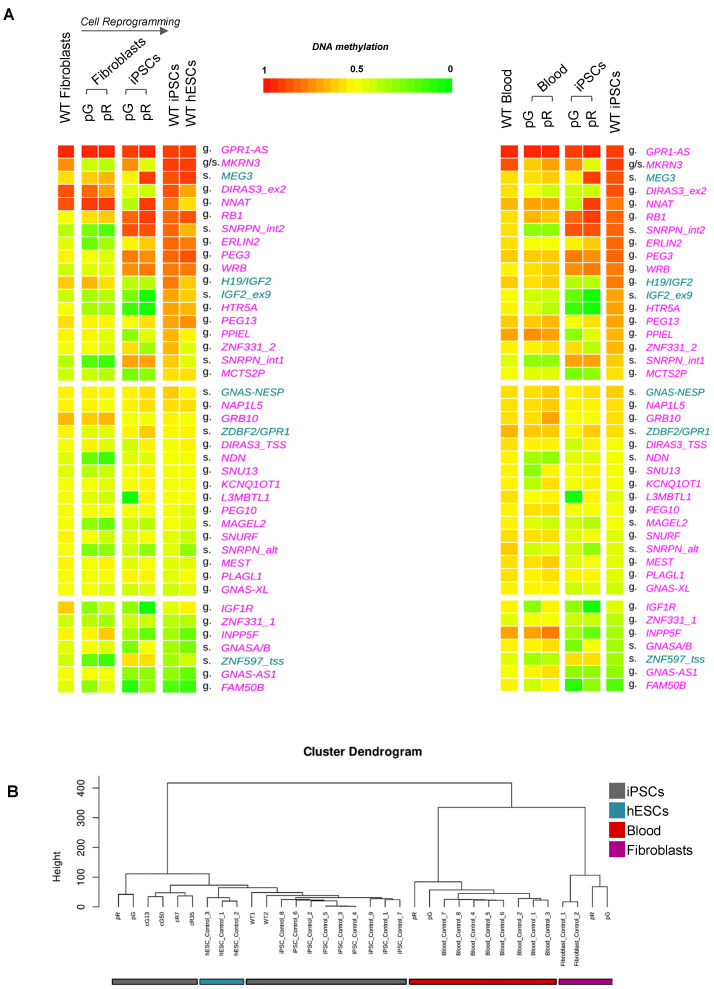
(**A**) Heatmaps showing the mean methylation level of iDMRs in ICF1 and control fibroblasts (left panel) or in ICF1 and control blood samples (right panel) compared to ICF1 and control iPSCs. Control hESC samples are also included in the heatmap (left panel) to compare the mean methylation level of iDMRs between different WT pluripotent stem cells. The iDMRs are displayed in the same order as shown in Figure 1A. The controls are displayed as median methylation value. (**B**) Sample clustering of control iPSCs, patient and DNMT3B-corrected iPSCs, control and patient fibroblasts, control hESCs, and control and patient blood DNA, based on DNA methylation data including all the shared probes. The height of the dendrogram is the distance between clusters.

**Figure 3 biomolecules-13-01717-f003:**
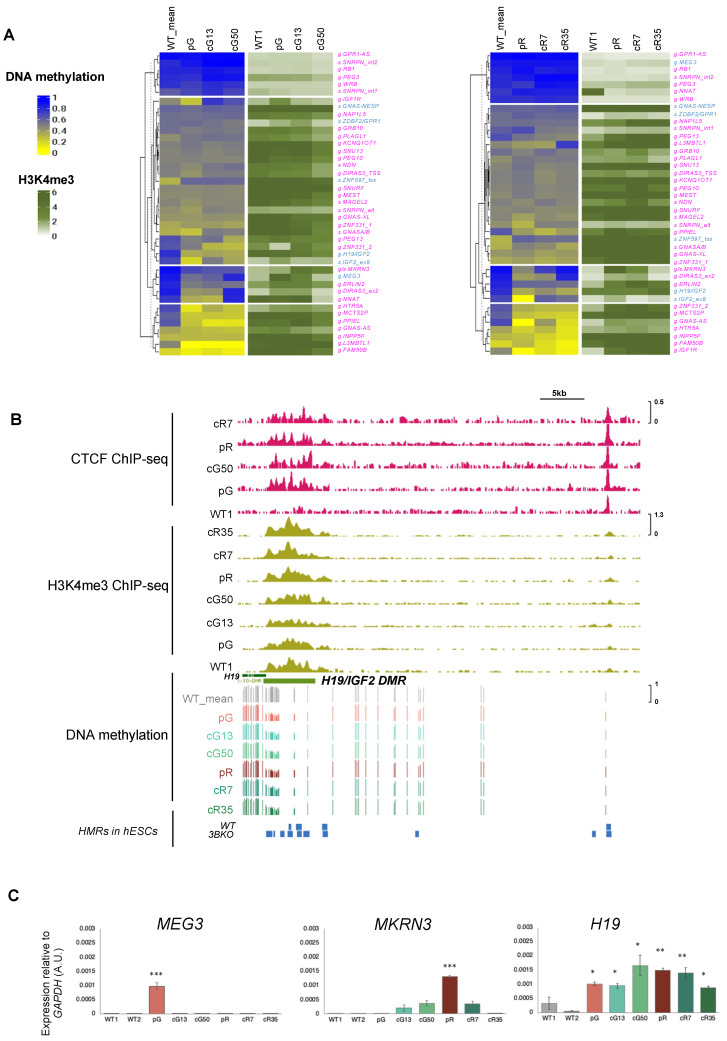
Altered H3K4me3 levels at hypomethylated imprinted DMRs. (**A**) A complex heatmap showing the integration of ChIP-seq data of H3K4me3 with DNA methylation levels at iDMRs (*n* = 41) in pR, cR7, cR35 (left panel) and pG, cG13, cG50 (right panel). The methylation levels are represented as β_ave_-values and H3K4me3 levels as FC = fold change (IP/input) of the ChIP-seq enrichment calculated across the corresponding DMRs. (**B**) Genome browser view of the *H19/IGF2* gDMR (light green bar below the GENCODE V43 gene tracks). The coverage tracks from the top to the bottom display the CTCF enrichment (pink) followed by H3K4me3 enrichment (light green) levels obtained from ChIP-seq experiments previously published [24] and the DNA methylation levels shown as β-values for all iPSCs (control, ICF1, and corrected iPSC clones) obtained in this study. The blue tracks underneath the DNA methylation levels illustrate the hypomethylated regions (HMRs) in wild-type (WT) and *DNMT3B* knockout (3BKO) human embryonic stem cells (hESCs; [11]). (**C**) RT-qPCR of the hypomethylated imprinted genes *MEG3*, *MKRN3*, and *H19* in WT, ICF1, and corrected iPSCs. Each bar represents the mean ± SD from independent triplicates (each amplified twice) of the relative expression compared to the expression of *GAPDH* in the same sample. Statistical analyses were performed using a one-tail two-sample Student’s *t*-test compared to WT: (*) *p*-value < 0.05, (**) *p*-value < 0.01, (***) *p*-value < 0.001.

## Data Availability

The raw and processed data related to this manuscript are available in the Gene Expression Omnibus (GEO) repository under accession number GSE198705.

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
