# Peer review of "ICF1-Syndrome-Associated DNMT3B Mutations Prevent De Novo Methylation at a Subset of Imprinted Loci during iPSC Reprogramming"

_biomolecules, 2023, doi:10.3390/biom13121717_

Round 1

Reviewer 1 Report

Comments and Suggestions for Authors

Verma et al studied ICF1 syndrome-associated DNMT3B mutations to prevent de novo methylation of several imprinted loci during iPSC reprogramming. This study used the unique platform of iPSCs carrying inactivated DNMT3B variants, and their corrected counterparts. The authors found that DNMT3B is responsible for de novo methylation of a subset of imprinted DMRs during iPSCs reprogramming, and the H3K4me3 mark is abnormally enriched in hypomethylated imprinted DMRs. Notably, the authors found that gene correction in ICF1 iPSCs cannot restore the methylation levels of iDMRs to those of control iPSCs.

Comments:

1. The whole study is based on previously obtained sequencing data, which has been published in Genome Res 2023 (PMID: 36828588) on a similar topic. The authors should discuss more about the novelty or difference of the current study.

2. Analytical data lack experimental validation, especially validation of methylation and ChIP-seq data.

Author Response

We thanks the three Reviewers for their valuable suggestions to improve the manuscripts. Please, see the attachment for the specific point by point response letter in which we addressed the comments of Reviewer 1.

Reviewer 2 Report

Comments and Suggestions for Authors

The study by Verma and co-workers explores the DNA methylation status of imprinted loci in hiPSCs with ICF1 syndrome-associated DNMT3B Mutations.

This is essentially a follow-up of their recent study published in Genome Research (Poondi Krishnan et al, 2023), with a focus on imprinted genes.

I have several concerns:

1. The authors claim that the I’ICF1 Syndrome-associated DNMT3B Mutations Prevent

de novo Methylation of Several Imprinted Loci During iPSC Reprogramming’.

Indeed, I do see in Figure 2A that some iDMRs are hypomethylated in patient lines compared to control hiPSCs lines, suggesting a role for DNMT3B. However, this does not seem to be corrected in their corrected lines, while we expect that reintroducing the enzyme will restore the DNAme levels. Can the authors comment on this?

By the way, the green dark and pale colours for both patient and corrected lines make them difficult to be distinguished in Fig2A.

In addition, the analysis shown in Fg2C should be done by gathering PG and PR lines and comparing them to the 4 corrected lines. Here it seems more of a line effect.

2. I suggest that the authors corroborate the data obtained with the Infinium Chips with another method that studies DNAme, including Sanger sequencing of bisulphite-treated DNA, at least for a couple of imprinted DMRs.

3. What is the impact on the level of expression of imprinted genes?

4. Figure 1 is essentially a different way of analysing what was published in the GR study. In my opinion, it does not help in a study focused on imprinted genes. Thus, I suggest to put it in supplementary data. By contrast, your suppl. Fig 4, which contains DNAme data on fibroblasts, is more informative/related to your present study and could be your new Figure 1.

Author Response

We thanks the three Reviewers for their valuable suggestions to improve the manuscripts. Please, see the attachment for the specific point by point response letter in which we addressed the comments of Reviewer 2.

Reviewer 3 Report

Comments and Suggestions for Authors

The research group led by the principal investigators has published several manuscripts related to ICF1 syndrome and epigenetic modifications. Following a line of research from the group, I only have a few comments about the study and its development. 

I would like more details about iPSC cells and CRISPCas9 editing. As written in the material and methods, the iPSCs in this study were taken from a previous study (Sagie et al., 2014). Understandably, in 2014, the group used retroviral vectors to reprogram iPSCs. However, I did not see the point of using this methodology since there are new methods to reprogram cells, like episomal. Retroviral vectors integrate their genetic material into the host cell's genome in their normal replication cycle. This integration can disrupt the host cell's genes, potentially leading to mutations, oncogene activation, or other genetic abnormalities. This is a significant concern because it may influence the DNMT3B activity in the iPSCs. In addition,  Retroviral vectors often introduce transgenes (genes from the virus) into the iPSCs, which can continue to be expressed in the reprogrammed cells. The presence of these transgenes may interfere with the normal functioning of the iPSCs and could be undesirable in specific applications. Then, before the authors proceed with the other assays, did you check if the exogenous genes continue to be expressed in the iPSCs cells? 

In studies related to epigenetic modification, it is more than recommended to maintain the same culture pattern, medium, and in which cell passage the research was carried out. Therefore, a simple cell change or passage can compromise the entire study. Consequently, I would like the authors to clarify in which cell passage the research was carried out and whether karyotypic analysis was previously carried out before proceeding with the methylation studies. Another detail for the female line of iPSCs: Did the authors check whether the X chromosome was active or inactive before moving with the other analyses?

The authors used CRISPR/Cas9-mediated editing to correct DNMT3B mutations. I would like the authors to describe in more detail how the cell editing was done and show how efficient the electroporation editing was.

Comments on the Quality of English Language

No comments

Author Response

We thanks the three Reviewers for their valuable suggestions to improve the manuscripts. Please, see the attachment for the specific point by point response letter in which we addressed the comments of Reviewer 3.

Round 2

Reviewer 1 Report

Comments and Suggestions for Authors

The authors have addressed all of my comments and I recommend to accept the paper.

Author Response

Thanks for your review.

Reviewer 2 Report

Comments and Suggestions for Authors

The authors have satisfactorily addressed my comments.

Author Response

Thanks for your review.